# Seed Dormancy and Seedling Ecophysiology Reveal the Ecological Amplitude of the Threatened Endemism *Picris willkommii* (Schultz Bip.) Nyman (Asteraceae)

**DOI:** 10.3390/plants11151981

**Published:** 2022-07-29

**Authors:** Manuel Fernández, Raúl Tapias

**Affiliations:** Department of Agroforestry Sciences, School of Engineering, University of Huelva, Avda. Fuerzas Armadas s/n, 21071 Huelva, Spain; rtapias@dcaf.uhu.es

**Keywords:** seed germination, plant emergence, seed heteromorphism, dormancy release, threatened species, temperature, light, substrate, plant ecophysiology

## Abstract

Plant communities can undergo drastic changes in their composition if the ecosystem is severely altered by human actions or climate change. These changes endanger any vulnerable species, mainly if it lives in a small area, as is the case of *Picris willkommii* (Schultz Bip.) Nyman. Therefore, it is essential to know how an ecosystem alteration could affect the seasonal pattern of the life cycle, seed production, germination time, as well as both plant emergence and development. During three consecutive years, the growth phenology and seed morpho-physiological traits of *Picris willkommii* were assessed, as well as the environmental factors that affect them (light, temperature, substrate). Under natural conditions, germination is in early autumn (15–25 °C air temperature), flowering is in spring, and seed maturation in late spring. The species produces two types of seeds differentiated in the degree of dormancy and other morpho–physiological traits, which contributes to the dispersal and spreading capacity; it prefers fine-textured limestone substrates with high N and P availability; it does not tolerate frosts below −5 °C; and it is able to acclimatize to changing environmental conditions, but there is a risk of being replaced by other more aggressive species. All of this is useful for species conservation programs.

## 1. Introduction

The seed is the higher plant perpetuation unit. It results from sexual reproduction and contains a developed plant embryo, which must remain alive during the period between seed maturation and seed germination, thus ensuring the next generation of the species [1]. After seed dispersal, germination and seedling emergence and establishment will determine the ecological amplitude and the geographical distribution of the species, because seedlings are much more vulnerable to biotic and abiotic unfavorable factors in the new location than the mother plant [2,3]. Compared to perennial species, annual species are even more vulnerable, since an extreme unfavorable factor between seed dispersal and plant establishment could wipe out all offspring. For all of the above, many species produce large quantities of seeds before plant death, capable of staying alive in the soil for more than a year (i.e., soil seed bank), combining survival strategies, such as the amount and the existence of diversity in the morpho–physiological characteristics of the seeds (size and shape of seeds, presence of dispersal structures, delayed seed dispersion, as well as seed dormancy) [4,5]. Therefore, the success of the next generation is largely determined by aspects such as the correct germination time in a favorable environment, seed size, and growth rate, which facilitates not only germination, but also plant establishment in the medium term [4,6,7].

The species can regulate the germination time by producing heterogeneous seeds as regards dormancy, either during maturation or after dispersal. Dormancy can be defined as a blocked state in which the seed does not germinate, even though there are favorable environmental conditions for it, and it can be determined by the seed coat or by the embryo itself [1,4,8]. It has been estimated that 50–95% of species have seeds with some type of dormancy at maturity (physical, physiological, morphological, morpho–physiological or combinational dormancy), and among these species, 50–75% have physiological dormancy [9,10] induced by the accumulation of water-soluble germination inhibitors or abscisic acid [4]. This means that seeds have inhibition mechanisms that must be broken before germination, to release the dormant state, which will depend on the dormancy type. Depending on the dormancy class, some techniques that simulate natural environmental stimuli, such as cold and warm stratification, dry storage, seed coat scarification, application of salinity or gibberellins, can cause the seeds to be released from dormancy [9,11].

Heterogeneity can affect not only physiological properties related to germination but also morphological characteristics, such as size, shape and color, as well as dispersal-related structures, such as wings, hooks, spines, pappus, etc. Imbert’s study [12] considered seed heteromorphism as the production of different kinds of seeds in a single individual and concluded that although it only affects less than 0.3% of Angiosperm species, it may have ecological implications for seed dispersal and soil seed bank maintenance. There are at least 218 species with that characteristic, of which 63% belong to Asteraceae [12]. Differentiation in this family mainly occurs because the central and peripheral achenes of a single capitulum (i.e., a flower head) show differences in their external appearance (size, shape, color and the presence of dispersal structures such as pappus or trichomes). Within the large Asteraceae family, the *Picris* genus encompasses just over 50 taxa distributed throughout Eurasia, Africa and Australia [13] that, apart from their ecological role, can be part of pastures for livestock, and even have medicinal use [14]. *Picris* species are morphologically variable herbs that produce mostly flowers with yellow ligules and achenes accompanied by a plumose pappus [13]. The achenes of the homocarpic species are morphologically and functionally identical and are accompanied by the plumose pappus for wind dispersal. By contrast, heterocarpic species have two types of achenes (seed dimorphism): the central ones, with the plumose pappus; and the peripheral ones, lacking pappus and enclosed by phyllaries that do not disperse but are fixed to the receptaculum and will fall in the proximity of the mother plant [6,12,13]. The natural evolution of the genus *Picris* has led to the appearance of taxa ecologically adapted to very different habitats, from coastal areas to high mountains. Those taxa adapted to the unpredictable arid environments of the Mediterranean region have usually undergone an evolutionary strategy that combines herbs with a semelparous life cycle and seed heteromorphism [13].

*Picris willkommii* (Schultz Bip.) Nyman (Asteraceae) is an endemic and stenochoric species that lives in grasslands and agricultural systems near the coast in the southwest of the Iberian Peninsula and grows under particular conditions of climate and soil [13,15,16]. It is threatened due to the fragmentation and reduction of its habitat by anthropogenic actions, such as the change in land use due to urban expansion and the transformation of traditional agricultural systems [15,16]. The plant produces a basal rosette of lobed leaves appressed to the soil during autumn–winter, developing orthotropic and plagiotropic floriferous stems 10–80 cm tall during spring, whose flowers produce two types of seeds [15] (Appendix A). In order to study management measures that help the *Picris willkommii* population recover, as well as the species conservation and its extrapolation to other threatened species, it is necessary to deepen our knowledge of aspects related to seed dormancy and environmental factors that affect seed germination and seedling establishment. The biology of seed dormancy cannot be understood without knowing the phenology and development of the plants and the environmental conditions of the germination time, and vice versa. Therefore, the present study aims to investigate the differences between the two seed types of *P. willkommii* with regard to dormancy and germination characteristics, as well as seedling development under different environmental conditions. To this end, the following aspects were addressed: (i) a brief approach to the growth cycle of plants and seed production in their natural environment; (ii) the morphology of achenes within a capitulum; (iii) the seed germination properties under different temperature and light conditions; (iv) the effect of pre-germination treatments that release seeds from their dormancy; and (v) the response of plants (growth, morphology, survival) to different growth conditions (substrate, nutrient and water supply, nursery cultivation and outplanting, and competition with other herbs).

## 2. Results

### 2.1. Brief Characteristics of the Natural Population and Its Ability to Produce Seeds

#### 2.1.1. Seasonal Variation of Natural Population Density and Seed Production

Taking into account all soil types as a whole, the seedlings emerged after the first autumn rains (Figure 1 and Appendix A), when the average temperature was of 18–22 °C (from 15.5 °C minimum and 26.5 °C maximum). The maximum plant recruitment was reached at the end of October. From that moment on, the emergence of new individuals was sharply reduced. Consequently, the population density remained more or less constant during the second part of autumn and the first half of winter, presenting slow vegetative growth, giving rise to plants with only basal leaves of about 5–15 cm in length. After the rise in temperatures from the second half of winter (beginning of February), vegetative growth accelerated, and reproductive growth began (development of flower stems). In the three years of study, no emergence of new plants was observed at the end of winter or in spring, except for some very sporadic individual ones in the gaps of the grassland, although spring temperatures and soil moisture allowed it. Plant density revealed significant differences between dates (*p* < 0.001) and between soil types (*p* = 0.001), but not for the *Date* × *Soil* interaction (*p* = 0.086). Soils on argillaceous blue marls (ABM) and from farmland (FL1) differed significantly from red sandy clay soils (RSC), with the ranking being ABM, FL1 ≥ FL2 ≥ RSC (see Section 4.1.1 for soil characteristics).

In some sampling points on ABM and FL1, up to 545 plants m^−2^ were recorded, but their number progressively decreased until it was less than 40 plants m^−2^ in May. However, on the opposite side, less than 5 plants m^−2^ came to be counted in plots over RSC. In any case, despite the great variation in the number of individuals between soil types and years, at the end of their reproductive cycle (first half of June) the average density converged to 5–10 plants m^−2^ (in a range of 1 to 50 plants m^−2^). This plant density was maintained until the plants’ senescence, which took place in mid-June.

Average daily temperatures of 15 °C (10–20 °C, minimum and maximum) at the end of winter favored the start of flowering, usually between February and early March. This stage had its peak in April and continued until the end of May. Likewise, the dissemination of the first seeds (the central ones), starting in mid-spring, coincided with 12–15 °C minimum daily temperatures and maximum temperatures of 25–30 °C, and lasted until the end of June, while the peripheral seeds remained attached to the mother plant.

Although an exhaustive study of plant distribution within the study area was not carried out, this distribution was not uniform, but rather followed a clumped pattern [17], concentrating the individuals in those areas with a more favorable microenvironment (soil fertility, moisture, etc.). In those places where the soil was covered by some plant species, mainly *Oxalis pes-caprae* L. and *Amaranthus blitoides* S. Wats.*,* but also *Chrysanthemum coronarium* L.*, Malva sylvestris* L., *Cynodon dactylon* (L.) Pers., *Scorpiurus sulcatus* L. y *Hordeum murinum* L., the presence of *Picris willkommii* was considerably reduced, even being nullified in the most severe cases (Appendix A). However, within a small area with homogeneous micro-environmental characteristics, the distribution of this species could be random.

The number of flowers per square meter depended both on the number of plants and their size. On the sampling date (end of April 2008), the average number of flowers was 67.2 ± 22.7 m^−2^ for ABM and FL1 soils, 17.3 ± 4.2 m^−2^ for FL2, and 7.0 ± 1.1 m^−2^ for RSC, with significant differences among soil types (*p* < 0.001). In the points with the highest plant density, up to 280 flowers m^−2^ were recorded. Under optimal growth conditions, a single individual reached up to 1.3 m in height, occupied 0.25 m^2^ of soil surface, and produced more than 400 flowers. In areas with more unfavorable environmental conditions, however, the average plant height did not reach 20 cm. On average, each flower usually produced 35–55 central seeds and 13 peripheral seeds. In extreme unfavorable conditions, plants of 3.5 cm in height, less than 1 mm in main root diameter and 1 cm^2^ of leaf area, which developed a single flower with only 5 peripheral ligules, were observed. It should be added that the number of flowers per plant was strongly correlated with the plant biomass and plant size (Figure 2).

#### 2.1.2. Physiological Activity (Gas Exchange)

*Picris wilkommii* showed high net photosynthetic rates (A) from autumn to mid-spring (A = 14 − 19 μmol m^−2^ s^−1^ of CO_2_), within the average range of the other species (A = 10 − 33 μmol m^−2^ s^−1^ of CO_2_) (Appendix A). Thereafter, as temperatures increased and rainfall decreased, net photosynthetic rates sharply decreased until the plants withered in mid-June. *Malva*, *Chrysanthemum* and *Diplotaxis* stood out for their higher spring photosynthetic rates. Regarding transpiration (E), the seasonal pattern was similar to that described for photosynthesis. *Picris wilkommii* behaved as a great water consumer, compared to the other species studied (E = 2.5 − 3.6 mmol m^−2^ s^−1^ of H_2_O), only surpassed by *Malva* in spring (Appendix A). Combining net photosynthesis and transpiration rates into water use efficiency (WUE = A/E) *Picris willkommii* showed one of the lowest efficiencies, at least during fall, winter, and early spring (Figure 3), only higher than *Hordeum murinum*. The values shown in Figure 3 and Appendix A correspond to the instantaneous gas exchange rates (just at the time of measurement) on sunny days with typical temperatures of those periods of the year. Variations in gas exchange rates would be expected on cloudy days or with extreme temperatures, especially for cold winter days.

### 2.2. Seed Analysis

#### 2.2.1. Seed Viability

The four collected seed batches showed high viability (Table 1) for both types of seeds (central and peripheral). Significant differences between seed types were obtained (*p* < 0.001), but not among collection years (*p* = 0.085) nor for the Seed type × Year interaction (*p* = 0.420). All central seeds whose external color (Munsell^®^) was between yellowish (5 Y 7/10) to light brown (5 YR 4/6 to 5 YR 3/4) were empty, while those showing a dark brown color (5 YR 3/2 or darker) were viable. Likewise, light brown peripheral seeds (7.5 YR 5/6 or lighter) were empty, while dark-colored ones (7.5 YR 4/4 or darker) were viable. It is worth noting the lower viability of the central seeds collected in 2007 compared to the other years studied, although not statistically significant.

#### 2.2.2. Effect of Environmental Conditions and Pre-Germination Treatments on Seed Germination

As can be seen in Table 2 and Table 3, the optimal germination temperature was close to 20 °C for both types of seeds (centrals and peripherals). The germination percentage (GP), germination energy (GE), and germination values (GV_Cz_, GV_DP_) were highly correlated with each other, but less so with the mean time to reach 50% of germinated seeds (MGT_50_). The mean germination time (MGT) could be determined for those treatments in which GP > 50%, that is, only for central seeds subjected to 15, 20 or 25 °C, for which MGT was 29.4 ± 13.0, 4.2 ± 1.2 and 27.3 ± 12.0 days, respectively. Regarding the comparison between seed types at temperatures close to the optimum (15–25 °C), the central ones showed a significantly higher germination capacity and shorter germination time than the peripheral ones.

The germination and pre-germination treatments tested to the central seeds (Table 4) revealed that only gibberellins (GA treatment) increased germination (up to almost 100% GP), and germination in the dark did not differ from that obtained under white light, both at 20 °C. However, the response of the peripheral seeds was different from the previous ones, due to their photosensitivity: germination under red light (GP = 18.7 ± 8.1%) and white light (GP = 19.5 ± 2.6%), differed significantly (*p* < 0.001) from those germinated in the dark (GP = 3.0 ± 1.0%) or under far-red light (GP = 2.7 ± 2.1%). The proportion of dormant peripheral seeds was around 80%. In no case did the fluctuating temperature (20/14 °C, day/night), nor did the previous warm or cold stratification improve the germination capacity of the seeds. MGT could only be determined for the central seeds subjected to 20 °C, GA and darkness treatments (4.2 ± 1.2, 1.8 ± 0.1 and 3.6 ± 0.8 days, respectively), and for the peripheral ones subjected to GA and KNO_3_ (23.1 ± 8.5 and 27.1 ± 10.0 days, respectively).

Finally, regarding the germination of seeds from the temporary aerial bank collected in autumn 2006, in Petri dishes under laboratory conditions, the germination (GP = 5.3 ± 3.0%) was significantly reduced (*p* < 0.001) with respect to the ones collected four months ago (GP = 19.5%) and stored at 2–3 °C in the dark. However, the application of gibberellins (GA) increased germination (GP = 50.0 ± 4.8%), without differing significantly from the KNO_3_ treatment, and very close to the germination percentages obtained for the gibberellin treatment applied to seeds collected in June (GP = 65%).

#### 2.2.3. Effect of the Substrate, the Time since the Seed Collection and the Seed Size on Germination

Regarding the substrate effect (natural soil) on seed germination, significant differences among substrates for germination percentage (GP) and germination values (Table 5) were obtained, although not very pronounced. In general terms, ABM, followed by FL1, stood out with higher values than the sandier soils (WSD, RSC and FL2).

The germination capacity (Table 6) was significantly lower for the seeds collected less than a month ago (2008), compared to those collected in previous years and stored in a dry and cool environment (2–3 °C) for a long time. In addition, the application of gibberellins to recently collected seeds (Table 7) significantly increased their germination potential.

Finally, the comparative analysis between seeds collected in 2005 and 2006 for seed mass did not reveal significant differences between collection years (*p* = 0.232), nor for the interaction Seed type × Year (*p* = 0.381). Significant differences, however, did exist between seed types (*p* < 0.001), with the mass of peripheral seeds being more than 4.5 times higher than that of central ones (Table 8). As a result, the 8-day-old seedlings developed from peripheral seeds were significantly larger than those from central ones (Table 9), 81% and 34% higher dry biomass and leaf area, respectively. In addition, the plants from central seeds developed thinner leaves, with a specific leaf area (SLA) 37% higher than those from peripheral seeds.

### 2.3. Effect of Environmental Variables on Plant Development

#### 2.3.1. Effect of Air Temperature and Light Radiation on Morpho–Physiological Traits

The maximum ranges of gas exchange rates corresponded to 20–25 °C for net photosynthesis (A) but 25–30 °C for transpiration (E). A direct proportional relationship was observed between temperature and water consumption (E) in the range 11–22 °C, while above or below this range E did not depend on temperature. This different behavior of A and E with respect to temperature led to water-use efficiency (WUE = A/E), showing its optimum value around 10 °C, while at temperatures close to the photosynthetic optimum (20–25 °C) or higher, this species had its minimum efficiency (Figure 4a). Chlorophyll a (Chl*a*) and b (Chl*b*) contents are shown in Table 10, with the ratio Chl*a*/Chl*b* = 4.1 ± 0.2 and SLA = 36.1 ± 4.4 m^2^ kg^−1^. 

In regards to cold tolerance, temperatures down to −2 °C did not cause any visible damage to the plants. At −3 °C, the youngest tissues and floriferous stems were damaged, but not the leaves of the basal rosette. Finally, at −5 °C, the damage extended to the basal rosette, although they did not completely die, in such a way that 50% of the plants recovered later.

Regarding now light radiation, the stems could adopt a horizontal or vertical growth habit depending on the light conditions in which they developed. A plant growing in a sunny, open space kept the basal leaves spread over the soil surface and developed almost horizontally growing stems (diageotropic) emerging from the center of the rosette, which contained very small leaves. Once they reached 20–40 cm in length, these stems turned upwards (orthotropic) to emit the flowers. However, a plant growing in high plant density or in a shaded area elongated the internodes at the base of the stem in such a way that there were no basal leaves, giving rise to a vertical main stem. The leaves at the base of the stem grew vertically and the leaves of the floriferous stems were larger than under sunny conditions (Figure 5).

This phenotypic plasticity was also shown in terms of leaf morphology, with leaves grown under high light intensity being more robust (lower SLA) than those grown under low light intensity (Figure 4b). Plants grown at light levels close to 50% of natural solar radiation (that is, about 850 μmol m^−2^ s^−1^ of photons) did not significantly change their leaf morphology, compared to those grown under full sunlight. Below 600 μmol m^−2^ s^−1^ of photons, SLA increased significantly. In no case were significant differences in SLA of the leaves developed on floriferous stems or on the basal rosette found (*p* = 0.357).

The maximum net photosynthetic rates (Amax), 21−25 μmol m^−2^ s^−1^ of CO_2_, were reached from 800 μmol m^−2^ s^−1^ of photons onwards. Half of the maximum net photosynthetic rate (½ of Amax) was obtained for 250 μmol m^−2^ s^−1^ of photons, the light compensation point around 5–10 μmol m^−2^ s^−1^ and an almost linear relationship between light intensity and transpiration was observed (Figure 4c). The maximum water use efficiency (WUE) was reached from 550 μmol m^−2^ s^−1^ of photons onwards, while the stabilization of the internal CO_2_ concentration (Ci) at light intensities above 500 μmol m^−2^ s^−1^ (Figure 4d). In addition, the plants only emitted by respiration between 0.25–0.65 μmol m^–2^ s^–1^ of CO_2_ at the measurement temperature (22 °C). 

#### 2.3.2. Effect of Substrate and Mineral Nutrients on Plant Growth

The plants size at the end of the trial (main stem diameter and dry mass) as well as the number of flowers and the biomass distribution are shown in Figure 6 and Table 11. It should be noted that, for the trial without additional fertilization, the treatments deficient in N and P, as well as those of natural soils, only reached 6–13% of the total dry mass obtained by the complete treatment, while treatments deficient in K, Mg, S, Ca, and Fe averaged 83–94%. In the case of P deficiency, some of the plants died without completing their fruiting process. Fertilizing natural substrates significantly increased plant growth compared to unfertilized ones. The form of nitrogen supplied, nitric or ammoniacal, did not induce differences in plant growth, but the amount supplied did, with the plants that received the greatest amount of fertilizer growing significantly more (250 ppm versus 126 ppm). 

#### 2.3.3. Effect of Planting Time under Field Conditions on Plant Development

This species satisfactorily withstood cultivation in the nursery and subsequent field planting (Appendix A). The planting date that required the fewest cultivation tasks was March, as the plot only had to be lightly weeded, without the need to apply irrigation. For this planting date, survival was 98%, with more than 50 flowers per plant on average. The April planting, however, needed periodic irrigation to promote rooting and growth, survival was 92% and an average of 40 flowers per plant was produced. Considering the plants planted in March and April as a whole, some few individuals produced more than 150 flowers. Finally, for the May planting, irrigation became absolutely essential, the survival did not exceed 80%, and the plant growth was much lower than that of the previous dates, since from a very early age, they developed stems and flowers. They did not reach 10 flowers per plant on average, and the percentage of fertile seeds was less than 50%. 

The plants flowered, bearing fruit with viable seeds that dispersed and germinated spontaneously the following fall, giving rise to a new generation. At the beginning of autumn, the density of the new population exceeded 10 plants of *Picris willkommii* per square meter. Even so, although *Picris willkommii* spontaneously regenerated quite well and managed to generate adult individuals and flowers, the new generation was seriously affected by competition from other plant species, mainly *Oxalis pes-caprae*, *Amaranthus blitoides*, *Malva* sp. and *Chrysanthemum* sp. (Appendix A).

## 3. Discussion

### 3.1. Brief Ecological Characteristics of the Natural Population and Seed Production

Considering the mosaic of existing soils in the natural range of this species, the highest abundance was found on limestone soils, with a loamy to clayey texture and the fertility degree as high as possible, such as ABM and FL1. The higher water retention and cation exchange capacity of fine-textured soils could be the reason that justifies it because they increase, respectively, the availability of water and mineral nutrients [18] and, consequently, plant abundance, growth, and the production of flowers and seeds [19,20,21]. The low water-use efficiency of *Picris willkommii* compared to the other grassland species could justify the preference for fine-textured substrates, with a higher water retention capacity. In addition, the low water-use efficiency would hamper its spread into the surrounding sandy soils, as well as its ability to withstand increasingly frequent prolonged drought periods. The practically null emergence of plants during summer, even though some rainy episodes were recorded every studied year, would indicate that high temperatures (daily mean ≥ 24 °C) do not promote germination of this species [22,23,24,25], and/or that the seeds undergo some type of dormancy when they mature from which they are not released until late summer [10,26,27,28,29]. Similarly, once the mild temperatures of early autumn had passed (daily mean around 20 °C), the low temperatures (daily mean < 15 °C) of late autumn and winter considerably reduced germination, as occurs with many other species of the Asteraceae family [6,30,31,32]. As the aforesaid drop in temperature coincides with the shortening of the photoperiod, it was not possible to differentiate which of these environmental stimuli could have the greatest effect on germination under natural field conditions. Likewise, it was not possible to differentiate whether the emerged plants came from central or peripheral seeds. The greater emergence recorded in the first year (autumn 2006) could be explained in part, either by the greater autumn rainfall (295 mm), compared to 192 mm in 2007 and 167 mm in 2008, and/or by a higher average temperature during September–October (21.9 °C), versus 20.8 and 19.2 °C, for 2007 and 2008, respectively. The mature plant density observed in the three years of study suggests a stability of population abundance.

Germination in early autumn and vegetative growth during the cold season can mean a strategy for the development of the basal rosette and the accumulation of energy reserves, taking advantage of the mild winter in the area, which will allow a more vigorous growth when spring arrives [26,27,31]. During none of the three years assessed was the emergence of new plants observed in late winter or early spring, except for some sporadic individual, and even though the temperatures and soil moisture levels allowed it, the competition between plants or the shading generated by the plants already installed, of the same or different species, could be the cause [19,33,34,35]. This competition not only affected the emergence of new plants, but also those that had previously emerged, since only 10–25% of the individuals that emerged in early autumn developed and produced seeds. In addition, it demonstrates once again that the next generation does not depend only on the number of seeds generated and the germination potential, but also on other biotic and abiotic factors interacting with each other [21,23,34].

Under natural conditions, flowering is eminently spring, as soon as the plants reach an adequate size, and the average maximum daily temperatures reach 20 °C. Subsequently, the seed maturation and dissemination takes place at the end of spring and early summer, coinciding with average maximum daily temperatures above 25 °C, which can be advanced or delayed depending on soil moisture. However, at this time, only the central seeds are dispersed (anemochory) since the peripheral seeds remain attached to the senescent plant and their dispersal will take place well into late summer or early autumn mainly by gravity (barochory), but also by animals (epizoochory) thanks to the small spines they have. In some cases, they can even be wind-dispersed along with portions of dead plants (unreleased) like a tumbleweed. The latter demonstrates a double dispersal strategy, as occurs with other Asteraceae species that present seed dimorphism [6,32,36]. Vegetative growth and flowering were earlier in the second study season (2007/08), which began around 20 January 2008, and could be due to a warmer January than usual (12.9 °C mean monthly temperature), compared to 11.3 °C and 10.2 °C for 2007 and 2009, respectively. In any case, although the dispersal of central seeds is promoted by rising temperatures in the second half of spring, the development of the reproductive organs is extended in time. The flowers appear and mature on the stems as they elongate, being able to coexist on the same stem flower buds; flowers with all their fully developed yellow ligules, dry flower heads still retaining the central achenes displaying their pappus and the peripheral achenes, and dry flower heads in which the central achenes have dispersed and only the peripheral ones remain (Appendix A). This extends the dispersal period and the new generation´s chances of success [37,38]. The results indicate that the regulation of flowering and seed maturation in this species depend more on temperature and soil moisture than on other factors, such as the photoperiod, but this would be an aspect to be studied in greater depth in future studies. Likewise, since this study did not delve into the processes related to seed development, from flower fertilization to seed maturation and dispersal, as well as the internal (genetic, physiological) and environmental factors that regulate each stage of the process until producing seeds with such differentiated characteristics in the same flower head, it would be interesting to undertake an investigation on the matter.

During the autumn and winter season, the photosynthetic rates of *Picris willkommii* were within the average range of the species belonging to the grassland, which demonstrates the good adaptation of *Picris willkommii* to this habitat. The relationship of temperature with photosynthesis and optimization of water use efficiency at low temperatures also support the hypothesis that species fitness increases when germination occurs in autumn. However, the higher photosynthetic rates and water use efficiency shown in spring by *Malva*, *Chrysanthemum* and *Diplotaxis* could indicate advantages in growth and competition for resources in the season of more vigorous growth (spring). This could be a disadvantage for *Picris willkommii* when the natural state of the ecosystem is altered. For its part, *Oxalis pes-caprae* did not justify its competitiveness by high net photosynthetic rates, but rather by the extraordinary density of shoots that it develops and the occupation of the soil surface since early autumn, which can lead to the replacement of other species in their first stages, including *Picris willkommii*. However, despite the fact that *Oxalis pes-caprae* begins to decrease its physiological activity from the beginning of spring in order to end its annual cycle in early May, there is not enough time for plants of the replaced species to emerge at that time and complete their reproductive cycle before the arrival of summer. Consequently, any alteration of the natural balance of the ecosystem, such as anthropogenic pressure, can lead to changes in the composition and abundance of the species of the new cohort [20,35,39] and, therefore, to the replacement of *Picris willkommii* by other plant species historically present in the area (e.g., some *Malva*, *Chrysanthemum* and *Diplotaxis* species) or by other more recent invaders (e.g., *Oxalis pes-caprae* and *Amaranthus blitoides*). The mere fact of plowing the soil, carried out during the outplanting trial, caused the appearance of abundant individuals of other opportunistic and/or invasive species of the plant community, such as those just mentioned, which surely benefited from soil disturbance.

### 3.2. Seed Viability and Germination

The number of seeds produced after the maturation period and their high viability (>85%) is indicative of the excellent adaptation of the species to the habitat, while providing enough propagules to guarantee its persistence in future years [19,40,41]. Visually, dark-colored seeds (dark brown to almost black) are usually viable.

According to behavior observed in its natural habitat, the optimum germination temperature in the laboratory was close to 20 °C for both seed types (central and peripheral). However, the optimum range for the central ones (15–25 °C) was slightly higher than for the peripheral ones (15–20 °C), with the central seeds germinating faster (MGT_50_ = 2.5 days) than the peripheral ones (MGT_50_ = 6.1 days). These optimal germination temperatures are within the temperature ranges shown by other herbaceous plants of the Asteraceae family (15–30 °C), depending on the climatic conditions of the habitat and the germination season of each species [6,24,25,30,32,36]. The statistical significance of the *Temperature × Seed type* interaction revealed that the central seeds were much more sensitive to temperature than the peripheral ones, germinating in 80% when the temperature was close to the optimum. However, up to 20% of central seeds still remained dormant, whose germination control could be internally regulated [10,27]. This leads us to think that most of the central seeds are released from dormancy at the end of summer and go into a quiescent state, being able to activate germination as soon as the environmental conditions are favorable for it. Similarly, the lower germination capacity of peripheral seeds would indicate a higher degree of innate or spontaneous dormancy [6,28,32,42], which would allow them to remain for longer in the soil seed bank, as well as germinate at slightly lower temperatures than the central seeds as fall progresses [42,43,44,45].

Likewise, the response of seeds to pre-germination treatments (cold and warm stratification), as well as to different germination conditions (fluctuating temperature 20/14 °C day/night, darkness, gibberellins or KNO_3_) confirms the double strategy of the species of producing two types of seeds that can increase the probability of success: on the one hand, high dispersal and colonization capacity by wind-dispersed seeds (centrals) without too many requirements for germination (not photosensitive, low dormancy percentage, high germination value); on the other hand, the conservation of a temporary aerial and edaphic bank of seeds (mainly peripherals), with slower germination as they overcome dormancy, preferring open gaps in the grassland [6,32,33]. These differentiated characteristics of both types of seeds can be included within the classic *r*/*K* selection theory [46], which tries to explain the relationship between the different reproductive strategies of species, or populations within a species, and the environment in which they live. According to this theory, central seeds tend to present an *r*-type strategy, while peripheral seeds a *K*-type strategy. In this case, seed dimorphism within the same population makes it possible to overcome different environmental conditions and resource availability.

The positive response of germination of both types of seeds to gibberellins reveals physiological endogenous dormancy at the time of maturation [10,23,29,32]. Additionally, the favorable response to potassium nitrate (KNO_3_) of peripheral seeds would indicate, apart from the release of physiological dormancy, a possible preference of the species for more fertile substrates for seed germination (at least with nitrates), and/or the presence of germination inhibitors that are eliminated by chemical oxidation by KNO_3_ [22,23]. The preference for ABM and FL1 substrates for germination, although not excessively different from the other soil types assessed, would justify this tendency to prefer slightly fertile substrates for germination. In both seed types, cold stratification (as it happens in winter under natural conditions), far from promoting germination, seems to induce secondary dormancy. This can be advantageous in avoiding or reducing spring germination (as field recruitment data showed) in which plant competition can cause high mortality. The same happens when the central seeds are subjected to warm stratification [22,44,45,47,48].

The low germination percentage of newly collected seeds in early summer (tested in July), and the activation of germination after the supply of gibberellins would indicate, as mentioned above, that most of the seeds mature in June under a non-deep innate dormancy state. This dormancy can be released by means of a physiological stimulant. However, the significant increase in germination parameters, repeating the test 3–4 months later (October), suggests that a certain percentage of seeds naturally achieves dormancy alleviation at the end of summer, perhaps induced by the high temperatures of the warm season or by the washing of water-soluble inhibitors, thanks to the rains of early autumn [10,27,28,29]. The seeds that remain in the temporary aerial bank (fixed to the dry mother plant) possibly extend the dormancy for a longer time because they suffer less washing of germination inhibitors than the seeds that fall to the ground. Nevertheless, despite the fact that many seeds are released from dormancy at the end of summer and are able to germinate just after the first autumn rains, a high percentage of dormant seeds still remains in the soil (soil seed bank), which may involve around a maximum of 70% of the dispersed peripheral seeds and 20% of the central ones. Seed losses due to predation and natural senescence must be deducted from this amount.

The larger size of the peripheral seeds compared to the central ones implies a greater storage of energy reserves, which makes it possible to stay longer in the soil seed bank than the central ones [19,25,49]. In turn, the larger initial size of seedlings produced from the peripheral seeds makes them more competitive against other species in the colonization of both, gaps and (undisturbed) grassland with high competition, than the central ones due to their greater initial growth and their smaller specific leaf area (SLA) [50]. A fast initial growth allows the development of the basal rosette in competitive grasslands and the accumulation of reserves for spring flowering. Additionally, the larger root systems that would develop are advantageous in gaps, as well as the smaller SLA due to the greater optimization of the loss of water (higher water use efficiency) and the respiration rates on a mass basis. These differences in size between the two types of seeds, and in size and SLA of seedlings, the lack of differences between collection years, together with the different degree of dormancy, the ability to remain in the soil seed bank and to compete in the first stages of life, as well as the preferred substrate types, can be taken into account for the in situ conservation plans through the collection and conservation of seeds for later sowing (for example, it can be collected mainly peripheral seeds at early summer, despite the fact that a large part of the central ones have dispersed), the management of the gaps size in the grassland, the control of competition from other species, etc.

Although the largest seeds (peripheral) are the most advantageous in recovering both scenarios, they require a greater energy investment in their production and have a shorter dispersal distance than the central one. The greater number of the latter allows locating and colonizing gaps despite the fact of suffering greater mortality. The existence of the two types of seeds gives the species a great adaptive capacity but implies having an internal control of resource allocation. Taking into account that there are 2.7–4.2 times more central than peripheral seeds in a flower head, and that the mass of a peripheral seed is 4.5 times greater than that of a central one, the plants allocate almost the same amount of energy and resources to produce both types of seeds, or favor the peripheral ones. For this reason, a certain energetic effort can lead to either a multitude of central seeds that will be scattered randomly in search of new niches and whose success is mainly based on the high number of seeds, or to a few peripheral seeds, more selective with respect to the conditions and the time of germination that provide a greater guarantee of success and a greater ability to stay in the soil seed bank. This reinforces the hypothesis of the double reproductive strategy of this species mentioned above (*r* and *K* strategies).

The maintenance of both types of seeds allows the species to survive in variable and unpredictable ecosystems. The different sizes and degrees of dormancy of the seeds also contribute to overcoming dry years, delays in the first autumn rains or long intervals between rains that cause high mortality in the cohorts. These events are increasingly frequent in the current climate change scenario. Seed dimorphism is not exclusive to this species within the *Picris* genus or the Asteraceae family [12,13], and not excessively frequent in the Asteraceae family since only 0.6% of the species present it [12], but it is very useful for annual species that live in unpredictable environments such as the Mediterranean climate, where they must explore all possible options to ensure a new cohort.

The causes of the differences in the degree of dormancy and the distribution of resources between the two types of seeds must be studied in depth at different levels, among others at the genetic level, to explain why the genes related to dormancy are expressed with greater intensity in the peripheral seeds. The greater allocation of resources per seed could be one of the reasons, but it is beyond the scope of this study.

### 3.3. Effect of Environmental Variables on Plant Development

Although the optimum temperature range for net photosynthetic rate (20–25 °C) occurs mainly in spring, the species also shows positive rates at temperatures of 10–15 °C and even lower. Therefore, it is not surprising that the basal rosette can grow and accumulate energy reserves during autumn and winter, on the many days with favorable temperatures at this time of year in the area where *Picris willkommii* lives. The drop in net photosynthesis begins at 25 °C, possibly due to an increase in respiration [51,52,53], while increasing transpiration and decreasing water use efficiency. This demonstrates its adaptation to the temperature regime of its natural range [52,54] and is related to the beginning of the senescence process in late spring. Temperatures below zero (specifically below –3 °C) seriously harm plants, so its potential expansion area is limited to frost-free areas, preferably coastal areas, but hardly inland.

Likewise, the leaf chlorophyll contents (Chl*a* and Chl*b*), as well as the Chl*a*/Chl*b* ratio, are within the normal range for a wide spectrum of species [55]. This demonstrates that this species properly regulates chlorophyll synthesis for adapting to growth conditions in such a way that it allows *Picris willkommii* plants to reach high net photosynthetic rates, up to 25 μmol m^–2^ s^–1^ of CO_2_ under optimal conditions. If to all this we add its phenotypic plasticity with respect to light intensity, revealed in the growth habit of stems and leaves (orthotropism, and diageotropism) and in the leaf structure (SLA), *Picris willkommii* has sufficient capacity to adapt to the growing conditions to be competitive, provided that some of the vital resources are not in short supply. The reduction in SLA under high light conditions implies a greater investment of biomass per unit of leaf area, which provides a high capacity for the conservation of the water status [50], thus increasing the water use efficiency at the time of optimizing assimilation by photosynthesis.

This species can survive and grow in non-excessively sunny environments, since its light compensation point is at 5–10 μmol m^–2^ s^–1^ of photons, and at 275 μmol m^–2^ s^–1^ half of the maximum photosynthetic rate is reached. However, in order to be able to compete successfully against other species, it may need higher light intensities. In addition, the fact that more than 800 μmol m^–2^ s^–1^ of photons are needed to reach maximum photosynthetic rates, the optimal WUE is achieved from 550 μmol m^–2^ s^–1^ of photons, and the stabilization of Ci occurs from 500 μmol m^–2^ s^–1^, it would indicate that above 500 μmol m^–2^ s^–1^ there are no limitations in the carboxylic function in chloroplasts, but between 500 and 800 μmol m^–2^ s^–1^, there could still be some stomatal limitation for photosynthesis [53].

Regarding the effect of the substrate on growth, the availability of water, N and P are the most limiting factors [20,21]. The scarce plants development on the unfertilized natural soil substrates (that is, the same order of magnitude as the treatments lacking N or P) shows the poverty in nutrients of the soils where *Picris willkommii* inhabits, and its tendency to present a clumped distribution, concentrating individuals on microsites with greater resources availability. 

### 3.4. Field Development after Outplanting

In order to restore or extend the species, late winter or early spring planting would be preferable to late spring, for survival reasons and because of the time needed for plant development, since high temperatures accelerate plant maturation and shorten the vegetative growth period, as mentioned above. It was also observed that the presence of pollinating insects was scarcer at the later planting date, which could affect the fertilization of the flowers [56]. All this could be the reason why, naturally, any seed that germinates in spring does not usually prosper or compete successfully in the grassland. Although data on winter or late fall plantings are not available, there is no reason to assume that they will not be successful.

However, although *Picris willkommii* reseeded itself spontaneously after outplanting and produced a new generation of adult individuals, the perpetuity of the species is seriously affected by the competition exerted by other plant species (mainly *Oxalis pes-caprae*, *Amaranthus blitoides*, *Malva* sp. and *Chrysanthemum* sp.). For this reason, the recovery of a species such as *Picris willkommii* in an altered environment will only be achieved if competition from other herbaceous plants is reduced by opening gaps in the grassland, until full establishment and the increase in the soil seed bank are reached, and, if necessary, implementing the availability of water and mineral nutrients [20,21].

## 4. Materials and Methods

### 4.1. Brief Ecological Characteristics of the Natural Population

#### 4.1.1. Seasonal Variation in Natural Population Density and Seed Production

The study area occupies about 600 ha in the southwest of the Iberian Peninsula, located between the following coordinates (UTM zone 29S; X_1_: 640480, Y_1_: 4122370; X_2_: 643703, Y_2_: 4122531; X_3_: 643621, Y_3_: 4120448; X_4_: 641106, Y_4_: 4120061) at an altitude from 4 to 60 m above sea level. The area has a Mediterranean climate, with dry and hot summers but mild winters, an average annual rainfall of 523 mm and an average annual temperature of 17.4 °C (Appendix A). This is the area with the greatest natural presence of the species, although some scattered individuals or groups can be found nearby [15,16]. Several types of soil outcrop in this area, such as soils on argillaceous blue marls (ABM), red sandy clay soils (RSC), and farmlands (FL). Five areas of 1200 m^2^ were randomly selected for sampling, with the condition of having unaltered or very little altered natural vegetation, mainly therophyte grasslands, except for those on farmlands: two of them on ABM, one on RSC and the other two on two types of farmland (FL1 and FL2). The physical–chemical properties of these soils are shown in Table 12. It should be noted that *Picris wilkommii* (Schultz Bip.) Nyman frequently inhabits ABM-type substrates, on some farmland such as FL1, less frequently on red sandy clay soil (RSC) and FL2, and very occasionally in forest clearings on white sandy dunes (WSD) and soils on slates (SLT). 

From December 2006 to June 2009, the number of *Picris wilkommii* plants was recorded on 13 different dates with periods of less than a month, as well as their phenological stage (i.e., basal rosette, presence of floriferous stems, presence of flowers, seed dispersal, and senescent plant). The measurements were concentrated during the vegetative period of each year, that is, from the beginning of autumn to the end of spring (see Figure 1), since in the summer the plants die, and the species remains in the form of seeds. Nine sampling points were randomly established within each study area (9 points × 5 areas = 45 points). Once established, they were marked with a stake in the NE corner, so that it was possible to measure the plots just in the same place during the rest of the study (Appendix A). At each sampling point, a square plot (0.5 m x 0.5 m) was established, with divisions every 10 cm to facilitate counting both the number of plants and the number of flowers. The air temperature and precipitation during the studied period were also recorded (Appendix A).

The relationship between plant size and the number of flowers was determined through a sample of 20 adult plants collected in the full flowering period (second week of April). Healthy plants of different sizes were selected. The total height, the main root diameter (measured at 0.5 cm depth) and the total and foliar dry biomass were measured.

#### 4.1.2. Physiological Activity (Gas Exchange)

The physiological activity of plants, through measurements of gas exchange (photosynthesis, transpiration), was carried out under field conditions during the third study season. Measurements were taken on seven different dates, from 8 November to 9 June (see Figure 3 and Appendix A), on sunny days and at mid-morning (between 10:30 and 12:30 local time), using a portable infrared gas analyzer (LCi, ADC^®^, Herts, UK) to which a broadleaf camera (PLC3, ADC^®^, Herts, UK) was attached. At all times, the photosynthetic solar radiation (400–700 nm) was between 1300 and 1600 μmol m^–2^ s^–1^ of photons. On each measurement date, four healthy plants per species belonging to nine plant species were measured. The selected species were *Picris willkommii* and another eight species of different genera abundant in the area that coexist with *Picris willkommi* in the grassland: *Malva sylvestris* L., *Chrysanthemum coronarium* L., *Diplotaxis virgata* (Cav.) DC., *Sonchus asper* (L.) Hill, *Anthyllis cytisoides* L., *Oxalis pes-caprae* L., *Scorpiurus sulcatus* L. and *Hordeum murinum* L.

#### 4.1.3. Data Analysis of Eco–Physiological Characteristics

Since the plant density (*Pd*, number of plants per m^2^) was evaluated in the same plots for several dates (Date) during three years in four soil types (Soil), our data structure resulted in repeated measurements for each plot. Therefore, a repeated measures ANOVA was applied [57], but each study season was analyzed separately. Hence, as the within-plot observations were autocorrelated, a model in which the plot (within-soil type) was considered, a random effect was used. Soil type (ABM, FL1, FL2, RSC) was included as a fixed effect. The model also included the pairwise interaction between the main effects as a fixed effect. It was tested in advance that the data met the assumptions of normality and sphericity. To evaluate the among soil comparisons, the Bonferroni test was used in order to differentiate the homogeneous groups. Significant differences were established at *p* < 0.05. Thus, the full model had the following structure for each study season: *Pd*_ijk_ = Date_i_ + Soil_j_ + (Date × Soil)_ij_ + ε_ijk_. The SPSS 19.0 software (IBM^®^ SPSS Statistics^®^) was used for data analysis.

### 4.2. Seed Analysis

Four seed batches were collected in the months of June of four consecutive years (2005, 2006, 2007 and 2008) inside the natural range of the species (more than 600 central seeds and more than 5000 peripheral seeds per year), and a smaller batch in October 2006 (about 400 peripheral seeds still persistent on the dry plants that remained standing). All of them were stored in hermetically sealed glass jars, in a dry and cold environment (2–3 °C) in the dark. During 2007 and 2008, seed viability and germination tests were performed. The experimental design differentiated between central and peripheral seeds (Appendix A), as well as between years of collection, as described later. After the germination tests, the germination parameters were determined, such as germination percentage at the end of the test (GP), germination energy (GE), mean germination time (MGT), germination value according to Czabator (GV_Cz_) and to Djavanshir and Pourbeik (GV_DP_), and mean time to reach 50% of germinated seeds (MGT_50_) [58].

#### 4.2.1. Seed Viability

The tetrazolium test was used to estimate seed viability [58]. For this, the seeds were previously hydrated in distilled water for 16 h (4 groups of 30 seeds per type and year of collection). Subsequently, two transverse cuts with a scalpel were given to each seed, one in the middle and another near the end where the radicle emerges in order to section the embryo. Once sectioned, they were immersed in a 0.5% 2,3,5-triphenyl tetrazolium chloride solution and kept in the dark for 6 h. After that, changes in color were observed in the cross-sections. Additionally, the color of the outer cover of 100 randomly selected seeds of each type (central and peripheral) was determined (Munsell® Color Chart) while they were sectioned in half to analyze whether the inner content had a seed or not.

#### 4.2.2. Effect of Environmental Conditions and Pre-Germination Treatments on Germination

From March to June 2007, three germination tests were carried out (5 weeks per test) using the seeds collected in June 2005 and 2006. For this, after separating the central seeds from the peripheral ones, they were disinfected with a mixture of fungicides (captan 1.5 g L^–1^ (Karnak 85®, 85% *w*/*w*) + benomyl 1 g L^–1^ (Afromyl® 50% *w*/*w*)) for 20 min and placed in Petri dishes containing sterilized filter paper moistened with distilled water (four sheets of paper per dish, previously disinfected with 0.5% sodium hypochlorite). The germination treatments (Table 13) were the following:First, a germination test at different temperatures under white light, at 10, 15, 20, 25 and 30 °C, in order to determine the optimal germination temperature.Second, germination and pre-germination treatments: once the optimal germination temperature was established in the previous test (close to 20 °C), germination tests were carried out at that temperature applying various germination and pre-germination treatments. The germination treatments were gibberellic acid (GA_3_) at 500 mg L^–1^, KNO_3_ at 0.2%, and type of light radiation, as well as variable temperature (20/14 °C, day/night). The pre-germination treatments were warm and cold stratification.Third, a germination test using peripheral seeds collected in October 2006, directly from the dry plant (the temporary air seed bank).

**Table 13 plants-11-01981-t013:** List of the different germination tests carried out according to the germination conditions, pre-germination treatments, type of seed and collection date; 11 h photoperiod.

Germination Conditions		Collection Date and Seed Type				
		June	2005	June	2006	October 2006
Temperature	Other Conditions	Central	Peripheral	Central	Peripheral	Peripheral
10 °C	WL + H_2_O_d_	X	X	X	X	
15 °C	WL + H_2_O_d_	X	X	X	X	
20 °C	WL + H_2_O_d_	X	X	X	X	X
25 °C	WL + H_2_O_d_	X	X	X	X	
30 °C	WL + H_2_O_d_	X	X	X	X	
	WL + GA_3_ (500 mg L^–1^)	X	X	X	X	X
	WL + KNO_3_ (0.2%)	X	X	X	X	X
	Darkness + H_2_O_d_	X	X	X	X	
20 °C	Red + H_2_O_d_		X		X	
	Far-Red + H_2_O_d_		X		X	
	WL + 20/14 °C + H_2_O_d_	X	X	X	X	
	Pre3°C + WL + H_2_O_d_	X	X	X	X	
	Pre30°C + WL + H_2_O_d_	X	X	X	X	
	Pre30/3°C + WL + H_2_O_d_	X	X	X	X	

Germination conditions: WL, white light (500 μmol m^–2^ s^–1^ of photons 400–700 nm); H_2_O_d_, distilled water; GA_3_, gibberellic acid 500 mg L^–1^ in H_2_O_d_; KNO_3_, 0.2% of KNO_3_ in H_2_O_d_; Darkness, under dark conditions; Red, under red light conditions (peak at wavelength 650 nm); Far-Red, under far-red light conditions (peak at wavelength 740 nm); 20/14 °C, fluctuating temperature day/night. Pre-germination treatments (i.e., before the beginning of the germination test): Pre3°C, moist cold stratification at 3 °C for 20 days; Pre30°C, dry warm stratification at 30 °C for 20 days; Pre30/3°C, warm stratification at 30 °C for 20 days followed by cold stratification at 3 °C for 20 days.

In all these germination tests, 25 seeds per Petri dish and 4 dishes were used per treatment, collection date and seed type (Appendix A). The photoperiod was set at 11 h, as this is the approximate value of early autumn, when the seeds supposedly begin to germinate in their natural habitat. The counts began on the second day after the beginning of the test and were repeated every 2–3 days until the end, on day 33.

#### 4.2.3. Effect of the Substrate, the Time since the Seed Collection and the Seed Size on Germination

From February to July 2008, new germination tests were carried out: (a)Effect of substrate: On the one hand, in February 2008, the effect on seed germination of the six most abundant soil types in the natural range of the species (Table 12) was tested, and an inert substrate was added as a control (commercial white perlite). This test was carried out at 20 °C, and only peripheral seeds collected in June 2005 were used (Appendix A): 200 seeds per treatment (4 pots of 50 seeds per treatment), moistened with distilled water, randomly distributed in the greenhouse (average temperature 20–22 °C and 10.5–11.5 h photoperiod during the test).(b)Effect of time since seed collection: On the other hand, in July 2008, the germination potential of the seeds collected and stored at 2–3 °C in the dark for different periods of time was evaluated. Therefore, the storage time of each seed batch was 3, 2, 1 years or 1 month, depending on whether they were collected in 2005, 2006, 2007 or 2008, respectively. Germination conditions were the same as those described above for the soil type test, but central and peripheral seeds were assessed (200 seeds per treatment and seed type).(c)Effect of seed size: Finally, on July 2008, the effect of the seed size (central and peripheral seeds) on the seedling size was tested. The fresh mass of the seeds (FM_s_), as well as the fresh mass (FM_p_), the dry mass (DM_p_) and the leaf area (LA_p_) of seedlings at 8 days after germination were assessed. Additionally, the dry mass of leaves (DM_l_) and the specific leaf area (SLA = LA_p_/DM_l_) were determined. This test was performed under laboratory conditions (20 °C, 11 h photoperiod, 4 Petri dishes per seed type).

#### 4.2.4. Data Analysis of Seed Viability and Germination

Seed viability and seed germination were analyzed by a three-way (4.2.2 first and second tests), two-way (4.2.1 and 4.2.3*b* tests) or one-way ANOVA (4.2.2 third, 4.2.3*a*, and 4.2.3*c* tests) by considering the fixed effects of seed type (ST—central or peripheral), germination/pre-germination treatments (TR—each of the treatment assessed) or seed collection date (Y–year), depending on the test. The model also included the pairwise and triple interactions between main effects as fixed effects. The models for three-, two- and one-way ANOVA were, respectively, as follows: *X*_ijkl_ = μ + ST_i_ + TR_j_ + Y_k_ + (ST × TR)_ij_ + (ST × Y)_ik_ + (TR × Y)_jk_ + (ST × TR × Y)_ijk_ + ε_ijkl_*X*_ijk_ = μ + ST_i_ + Y_j_ + (ST × Y)_ij_ + ε_ijk_*X*_ij_ = μ + ST_i_ + ε_ij_; or *X*_ij_ = μ + TR_i_ + ε_ij_

Normality and homoscedasticity were previously verified. Significant differences were established at *p* < 0.05. To evaluate the among-seed type/treatment and year comparisons, the Tukey HSD test was used. The SPSS 19.0 software (IBM^®^ SPSS Statistics^®^) was used for data analysis.

### 4.3. Effect of Environmental Variables on Plant Development

#### 4.3.1. Air Temperature and Light Radiation

The effect of temperature was assessed on plants grown in a greenhouse under 10–20 °C of daily temperatures (minimum–maximum) and a 11 h photoperiod, similar conditions to those that occur in its natural range during February and March (Appendix A), a time of high vegetative growth rates. In a sample of four plants, randomly chosen from 50, gas exchange (photosynthesis and transpiration) was measured with an infrared gas analyzer (LCi, ADC®, Herts, UK), under saturating light (PAR = 1200 μmol m^–2^ s^–1^ of photons, 400–700 nm) and at temperatures between 4 °C and 31 °C (range of daytime temperatures that cover practically the entirety of its active vegetative period in its natural habitat). Next, the specific leaf area (SLA) and leaf chlorophyll content [59] were determined for these plants. Regarding the effect of light radiation, a random sample of four plants was used to study the light response curves [60], measured at 22 °C with the aforementioned gas analyzer.

The effect of sub-zero temperatures (that is, cold tolerance), was evaluated in January 2008 by subjecting the whole plants in their pots to four freezing temperatures (–1, –2, –3 and –5 °C). The cold tolerance test was performed by simulating frosts in a freezer that had a temperature controller [61]. Briefly, when the whole plants were inside the freezer, the temperature was lowered at a rate of 3 °C h^–1^ from room temperature to the minimum programmed in each test. Once this minimum temperature was reached, it was kept for 3 h and then raised (5 °C h^–1^) to room temperature. Six plants per temperature were used, grown in pots in the nursery, outdoors with similar temperature to that of the natural area (Appendix A). After applying the thermal shock, the plants were taken to a greenhouse (temperature range 20/10 °C, day/night; 11 h of photoperiod) and watered as needed. The damage was visually observed one week after the application of the freezing temperature, by taking into account the proportion of dead parts (leaves and stems) in the aerial part of the plant.

Additionally, the specific leaf area (SLA) of plants grown in the same nursery but under four different light intensities (110, 375, 875 and 1500 μmol m^–2^ s^–1^ of photons) was determined. The different light intensities were achieved by means of layers of white polypropylene mesh, except for the maximum light radiation treatment (1500 μmol m^–2^ s^–1^) that corresponded to outdoor cultivation, without mesh.

#### 4.3.2. Substrate and Mineral Nutrients

In order to analyze the effect of soil type on plant growth, soil samples were taken from the surface layer of the edaphic profile (0–20 cm), once the litter layer had been removed, in the natural area where *Picris willkommii* lives. This trial had 15 treatments: 6 soil types (FL1, FL2, SLT, ABM, RSC and WSD, see Table 12) to which another 9 treatments were added, consisting of an inert substrate fertilized with nutrient solutions that differed from each other in the content of one of the macronutrients (N, P, K, Mg, S, Ca and Mg) or a micronutrient (Fe). Each treatment consisted of six repetitions of a single plants growing in 300 cm^3^ pots randomly arranged in the nursery. Previously, five seeds per pot were germinated on 12 February 2008, to ensure the availability of plants in all pots. After 2 weeks, once germinated, two seedlings per pot of similar size were left. Later, after 5 weeks, one of the two plants from each pot was removed and used to estimate the initial dry mass of the plants. In this way, only one plant remained per pot until the end of the trial, 105 days after the start. It was watered as needed throughout the trial. The 15 treatments are summarized as follows (see also Figure 6 and Table 11):

Six treatments corresponding to unfertilized natural soils (FL1, FL2, SLT, ABM, RSC and WSD).Nine mineral nutrition treatments on inert substrate (mixture of perlite: unfertilized *Sphagnum* peat: washed river sand (3:1:1 volume)). They were fertilized once a week with their corresponding nutrient solutions, whose pH were 5.5–5.7, with the following treatments:
-Complete treatment: nutrient solution composed of 126, 71 and 90 mg/L of N, P and K, respectively, also containing all other macro and micronutrients in balanced proportions. Nitrogen was added as N-NO_3_.-A treatment reduced in N (1/10N): the amount of nitrogen was reduced to a tenth of that provided in the complete treatment, maintaining the contents of the other nutrients in the same quantities as in the complete treatment.-Seven treatments, each one lacking a mineral nutrient (–N, –P, –K, –Ca, –Mg, –S, –Fe). Only the indicated nutrient was eliminated for each of them, providing the other nutrients in the same amount that in the complete treatment.

In addition, another parallel trial was established in which additional fertilization was applied to seven of the previous treatments (the six natural soils and one artificial substrate; see Figure 6 and Table 11). The same number of plants per treatment and the same type of pot and previous nursery cultivation conditions were used. Nutrients were supplied to all of them in similar proportions to the complete treatment but with double the amount (that is, 252, 142 and 180 mg L^–1^ of N, P and K), but in which the nitrogen was applied in nitric form (50% of N–NO_3_) and ammoniacal (50% of N–NH_4_), and the pH was around 5.8. The treatments were called FL1_F_, FL2_F_, SLT_F_, ABM_F_, RSC_F_, WSD_F_ and CompleteF.

#### 4.3.3. Plant Survival According Planting Time in the Field

During the 2007/008 study season, plants were grown in the nursery (Appendix A), in 300 cm^3^ pots filled with the inert substrate described above. They were planted in the field on FL1-type substrate. The land was previously plowed, sprinkler irrigation was installed and was divided into three parts, one for each planting date. Planting was carried out on three different dates (mid-March, mid-April and mid-May), 150 plants per date. Plant survival and the number of flowers per individual were recorded, as well as an estimation of the percentage of fertile seeds per flower (according to the color of the cover and whether they were empty). Likewise, an estimation of the density of emerged individuals was carried out in the following autumn, in mid-October.

#### 4.3.4. Data Analysis on Plant Response to the Environment

The data were treated statistically by one-way ANOVA following a general Linear Model (*X*_ij_ = μ + TR_i_ + ε_ij_), previously carrying out the pertinent verifications of normality and homoscedasticity. The main effects (temperature, light radiation, mineral nutrition, etc.) were considered fixed. Significant differences were established at *p* < 0.05. To evaluate the among-treatment comparisons the Tukey HSD test was used. The SPSS 19.0 software (IBM^®^ SPSS Statistics^®^) was used for data analysis. For the field planting trial, since no repetitions were made and three different planting points were used, the results are shown descriptively, without applying a complex statistical analysis.

## 5. Conclusions

-The timing of plant emergence (early autumn), flowering (spring but extended in time) and seed maturation and dispersal (late spring and extended in time) of *Picris willkommii* under natural conditions were determined, as well as the environmental conditions (temperature, soil moisture) in which each phase takes place.-The species presents dimorphism in seeds (central and peripheral), which show different characteristics, not only morphological, related to the size and dispersion medium, but also physiological (germination, dormancy, and photosensibility). This may indicate an evolutionary strategy in order to increase chances of colonization and survival.-Evidence of physiological endogenous dormancy was found in the seeds at the end of the plant’s maturation period, when they are ready for dispersal.-At the end of summer, coinciding with the first autumn rains and the drop in temperatures, the plants emerge, which will develop vegetative growth during autumn-winter and will finish the annual reproductive cycle between late spring and the beginning of the following summer.-The optimum temperature for seed germination of this species is close to 20 °C for both types of seeds (central and peripheral), although with slight differences between them.-Ripe seeds collected in the field in early summer have high viability and can be stored for a long period of time in a dry and cold environment (2–3 °C).-Within its natural range, the species prefers limestone soils, with active calcium, clayey or clay loam, and with a certain degree of fertility, mainly with N and P availability.-The alteration of the ecosystem can lead to the replacement of *Picris willkommii* by other more aggressive species, both native and invasive.-Sexual propagation, nursery cultivation and field planting is technically possible. In this case, it is recommended to control other competing herbaceous plants in the early stages of the life cycle.-Not only the seeds but also the vegetative organs (leaves and stems) are highly photosensitive. It is a species that tolerates shade but prefers exposure to light to optimize its growth and compete with a greater chance of success.-Winter temperatures below –5 °C harm the normal development of plants and reduce their ability to compete for resources.-The results obtained are useful for the preservation of the species against the alteration of the ecosystem for any cause, including climate change, as well as being useful for the implementation of conservation and recovery programs for threatened species.

## Figures and Tables

**Figure 1 plants-11-01981-f001:**
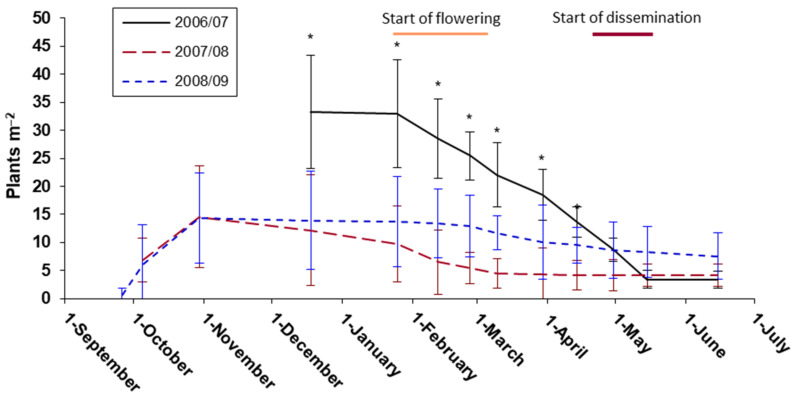
Average population density throughout the three studied periods. Asterisks indicate those dates in which significant differences between years were obtained (*p* < 0.05). Bars are standard errors. The periods of beginning of flowering and seed dispersal are also indicated.

**Figure 2 plants-11-01981-f002:**
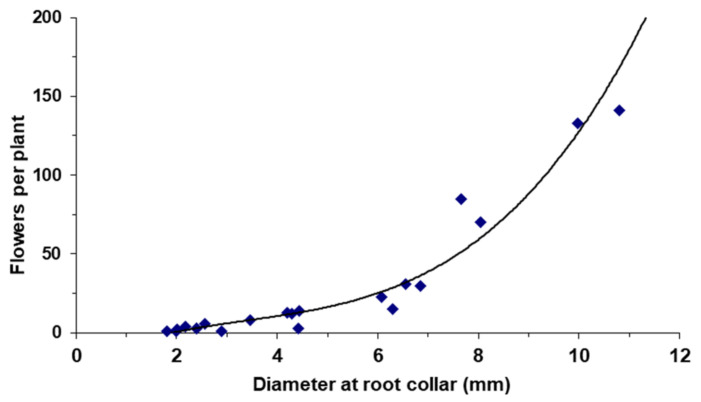
Relationship between root diameter (RD), measured 0.5 cm below the root collar, and the number of flowers produced per plant (PF). PF = 0.312 RD^3^ − 3.181 RD^2^ + 15.342 RD − 19.825 (R^2^ = 0.986; *p* < 0.001).

**Figure 3 plants-11-01981-f003:**
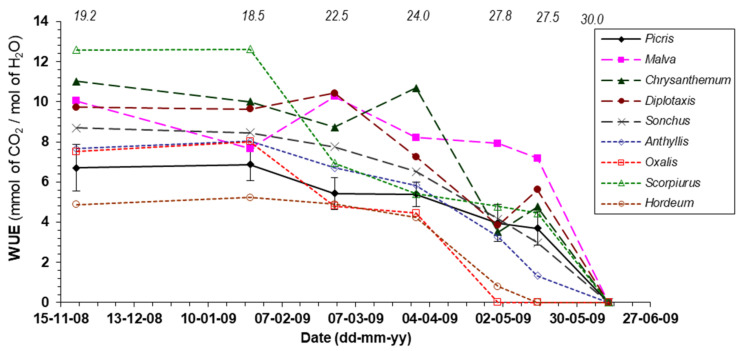
Seasonal pattern of instantaneous water-use efficiency (WUE) for nine plant species on sunny days. The numbers above indicate the ambient temperature at the measurement time (°C). The differences among species were significant in all the measurement dates (*p* < 0.001), except in June due to plant senescence. Species: *Picris willkommii* (Schultz Bip.) Nyman, *Malva sylvestris* L., *Chrysanthemum coronarium* L., *Diplotaxis virgata* (Cav.) DC., *Sonchus asper* (L.) Hill, *Anthyllis cytisoides* L., *Oxalis pes-caprae* L., *Scorpiurus sulcatus* L. y *Hordeum murinum* L. In order to clarify the figure, only the error bars corresponding to *Picris willkommii* are represented.

**Figure 4 plants-11-01981-f004:**
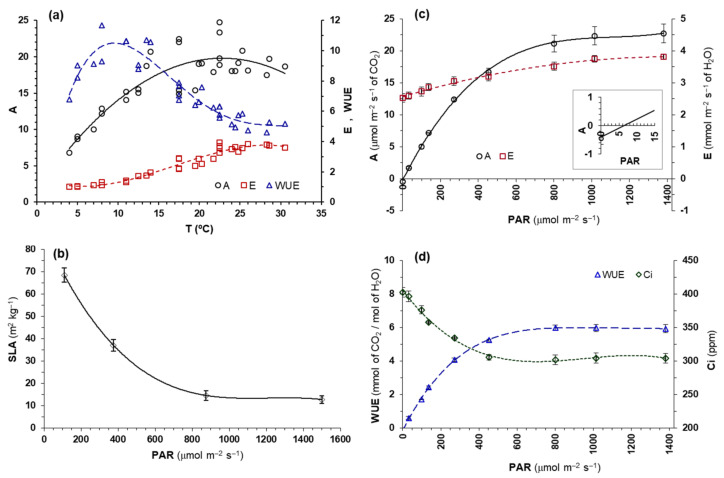
(**a**) Net photosynthetic rate (A, μmol m^−2^ s^−1^ of CO_2_), transpiration rate (E, mmol m^−2^ s^−1^ of H_2_O) and water use efficiency (EUA, mmol of CO_2_/mol of H_2_O) of *Picris willkommii* as a function of ambient temperature, measured under saturating light. A = −0.035 T^2^ + 1.601 T + 1.543 (R^2^ = 0.754). E = −0.0005 T^3^ + 0.024 T^2^ − 0.219 T – 1.607 (R^2^ = 0.947). WUE = −0.0001 T^4^ + 0.009 T^3^ – 0.285 T^2^ + 3.308 T – 2.154 (R^2^ = 0.892). (**b**) Relationship between specific leaf area (SLA) and light radiation (PAR, μmol m^−2^ s^−1^ of photons in the range 400−700 nm wavelength). The bars indicate the standard error. Sample size = 22 plants per light treatment. (**c**) Net photosynthetic rate (A) and transpiration rate (E), as well as (**d**) water use efficiency (WUE) and internal CO_2_ concentration of the leaf mesophyll (Ci), of *Picris willkommii* measured at 22 °C and different light intensities. A = 2 × 10^−^^8^ x^3^ − 5 × 10^−^^5^ x^2^ + 0.058 x − 0.155 (R^2^ = 0.999); E = −6×10^−^^7^ x^2^ + 0.002 x + 2.571 (R^2^ = 0.991); WUE = −4 × 10^−^^12^ x^4^ + 2 × 10^−^^8^ x^3^ – 3 × 10^−^^5^ x^2^ + 0.022 x – 0.130 (R^2^ = 0.999); Ci = −1 × 10^−^^7^ x^3^ + 4 × 10^−^^4^ x^2^ − 0.375 x + 405 (R^2^ = 0.995).

**Figure 5 plants-11-01981-f005:**
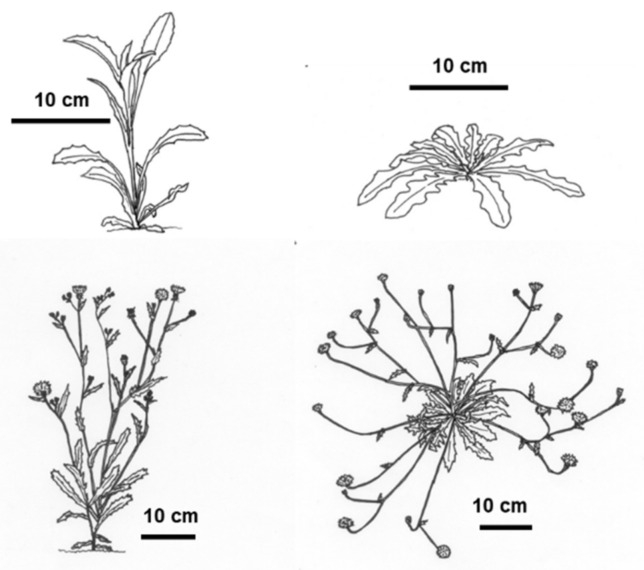
Phenotype of plants grown in high competition (**Left**) or in open space (**Right**), both in juvenile (**Top**) and mature (**Bottom**) stages.

**Figure 6 plants-11-01981-f006:**
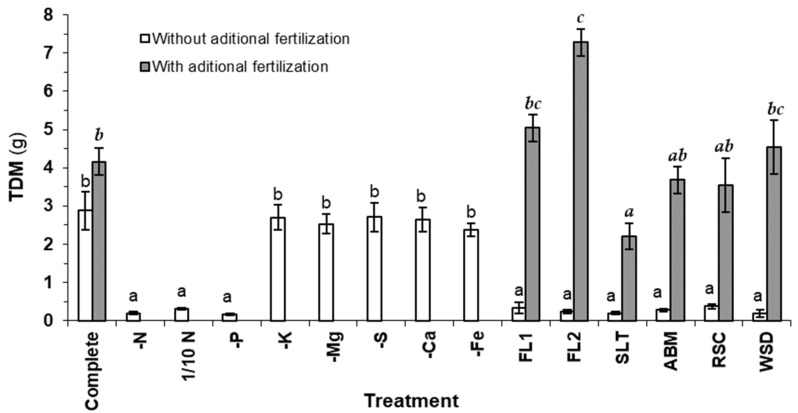
Total dry biomass (TDM) per treatment at the end of the mineral nutrition and substrate trials (mean ± SE), both without and with additional fertilization. Different letters on each bar indicate significant differences between treatments within each trial (*p* < 0.001).

**Table 1 plants-11-01981-t001:** Viability percentage of seeds (mean ± SE) according to the tetrazolium test.

Collected Year	Seed Type	
	Central	Peripheral
2005	97.2 ± 1.1	77.0 ± 2.8
2006	96.2 ± 1.5	70.7 ± 3.0
2007	85.5 ± 2.3	72.3 ± 2.2
2008	98.7 ± 1.0	80.1 ± 3.1

**Table 2 plants-11-01981-t002:** Mean values (± SE) and statistical significance of the germination parameters assessed at different temperatures, in the lab under white light. Averages for seeds collected in June 2005 and June 2006 are shown as a whole.

Seed Type	Temperature (°C)	GP (%)	GE (%)	GV_Cz_	MGT_50_ (Days)
	10	13.3 ± 3.2 c	7.8 ± 1.9 c	0.6 ± 0.3 c	10.0 ± 4.0 a
	15	51.0 ± 7.7 b	41.0 ± 7.2 a	13.6 ± 4.3 b	3.5 ± 0.3 a
Central	20	80.0 ± 6.3 a	50.0 ± 6.2 a	42.2 ± 7.3 a	2.5 ± 0.3 a
	25	56.0 ± 12.6 b	27.0 ± 5.3 b	19.9 ± 9.4 b	7.7 ± 3.5 a
	30	13.0 ± 5.6 c	9.5 ± 4.9 c	0.3 ± 0.2 c	12.7 ± 5.3 a
	10	3.3 ± 2.0 c	3.3 ± 2.0 c	0.1 ± 0.1 c	8.9 ± 4.0 a
	15	17.0 ± 2.1 ab	9.0 ± 1.2 a	0.8 ± 0.2 b	9.0 ± 2.0 a
Peripheral	20	19.5 ± 2.6 a	10.0 ± 2.2 a	2.1 ± 0.6 a	6.1 ± 2.0 a
	25	7.5 ± 2.9 bc	6.0 ± 1.7 b	0.4 ± 0.2 bc	6.6 ± 3.0 a
	30	8.8 ± 7.2 bc	8.8 ± 7.2 ab	0.3 ± 0.2 bc	15.4 ± 5.3 a
	Temperature	<0.001	<0.001	<0.001	0.061
	Seed type	<0.001	<0.001	<0.001	0.003
Significance	Year	0.100	0.090	0.101	0.457
level (*p*)	T × S	<0.001	<0.001	<0.001	0.349
	T × Y	0.004	0.170	<0.001	0.370
	S × Y	<0.001	0.003	<0.001	0.563
	T × S × A	0.151	0.245	0.101	0.269

Germination percentage (GP), germination energy (GE), germination value after Czabator (GV_Cz_), and mean time to reach 50% of germinated seeds (MGT_50_). Different letters in the same column indicate significant differences among temperature treatments for the same type of seeds. Germination conditions: white light (500 μmol m^−2^ s^−1^ of photons 400–700 nm) and moistened with distilled water.

**Table 3 plants-11-01981-t003:** Linear correlation coefficient (r, Pearson) between the germination parameters. Sample size: N = 62.

	GP	GE	GV_Cz_	GV_DP_	MGT_50_
GP	1.000				
GE	0.908 **	1.000			
GV_Cz_	0.914 **	0.828 **	1.000		
GV_DP_	0.927 **	0.812 **	0.991 **	1.000	
MGT_50_	−0.399 *	−0.400 *	−0.233	−0.233	1.000

Germination percentage (GP), germination energy (GE), germination values after Czabator (GV_Cz_) and Djavanshir and Pourbeik (VG_DP_), and mean time to reach 50% of germinated seeds (MGT_50_). **, *: correlations are significant at the 0.001 and 0.01 level, respectively.

**Table 4 plants-11-01981-t004:** Mean values (± SE) and statistical significance of the germination parameters assessed at different germination and pre-germination treatments, in the lab and under white light, except for the treatment of darkness. Averages for seeds collected in June 2005 and June 2006 are shown as a whole.

Seed Type	Treatment	GP (%)	GE (%)	GV_Cz_	MGT_50_ (Days)
	20 °C	80.0 ± 6.3 b	50.0 ± 6.2 b	42.2 ± 7.3 b	2.5 ± 0.3 a
	20/14 °C	37.0 ± 6.1 c	18.5 ± 7.9 cd	11.7 ± 7.9 c	9.0 ± 4.4 a
	GA	97.0 ± 1.9 a	81.9 ± 4.7 a	83.0 ± 6.2 a	1.8 ± 0.1 a
Central	KNO_3_	45.1 ± 11.5 c	26.0 ± 9.0 c	16.7 ± 8.7 c	2.4 ± 0.5 a
	Pre3°C	5.6 ± 2.0 e	12.5 ± 6.0 de	0.4 ± 0.3 d	5.5 ± 3.5 a
	Pre30°C	22.6 ± 4.0 d	14.0 ± 1.5 d	1.8 ± 0.5 d	2.8 ± 0.7 a
	Pre30/3°C	14.1 ± 3.0 e	6.0 ± 1.5 e	1.5 ± 0.1 d	1.6 ± 0.5 a
	Darkness	75.2 ± 7.8 b	50.0 ± 7.0 b	46.7 ± 10.5 b	1.8 ± 0.5 a
	20 °C	19.5 ± 2.6 b	10.0 ± 2.2 b	2.1 ± 0.6 c	6.1 ± 2.0 a
	20/14 °C	16.0 ± 2.7 bc	9.8 ± 1.6 bc	1.1 ± 0.5 c	6.1 ± 1.6 a
	GA	65.0 ± 6.6 a	30.0 ± 4.2 a	12.3 ± 2.6 a	6.5 ± 1.5 a
Peripheral	KNO_3_	54.7 ± 10.9 a	25.0 ± 8.5 a	5.6 ± 1.5 b	8.6 ± 2.9 a
	Pre3°C	10.8 ± 1.4 c	8.5 ± 3.0 c	1.8 ± 0.8 c	2.3 ± 0.4 a
	Pre30°C	19.6 ± 7.0 bc	11.0 ± 3.0 b	1.7 ± 0.7 c	4.6 ± 0.2 a
	Pre30/3°C	10.8 ± 1.4 c	5.8 ± 1.2 c	1.2 ± 0.3 c	4.0 ± 1.5 a
	Darkness	3.0 ± 1.0 d	2.8 ± 0.9 d	0.1 ± 0.1 d	4.9 ± 2.0 a
	Treatment	<0.001	<0.001	<0.001	0.054
	Seed type	<0.001	<0.001	<0.001	0.153
Significance	Year	0.062	0.053	0.098	0.645
level (*p*)	T × S	<0.001	0.008	<0.001	0.219
	T × Y	0.069	0.159	0.830	0.612
	S × Y	<0.001	0.004	0.011	0.475
	T × S × Y	0.806	0.765	0.745	0.823

Germination percentage (GP), germination energy (GE), germination value after Czabator (GV_Cz_), and mean time to reach 50% of germinated seeds (MGT_50_). Different letters in the same column indicate significant differences among treatments for the same type of seeds. Germination conditions: GA, gibberellic acid 500 mg L^−1^ in H_2_O_d_; KNO_3_, 0.2% of KNO_3_ in H_2_O_d_; Darkness, under dark conditions; 20/14 °C, fluctuating temperature day/night. Pre-germination treatments (i.e., before the beginning of the germination test): Pre3°C, moist cold stratification at 3 °C for 20 days; Pre30°C, dry warm stratification at 30 °C for 20 days; Pre30/3°C, warm stratification at 30 °C for 20 days followed by cold stratification at 3 °C for 20 days.

**Table 5 plants-11-01981-t005:** Mean values (±SE) and significance level (*p*) of the germination parameters assessed on different substrates (soil types), in the greenhouse and under natural light, using peripheral seeds collected in June 2005.

Seed Type	Substrate (Soil Type)	GP (%)	GE (%)	GV_Cz_	MGT_50_ (Days)
	Perlite	29.5 ± 3.2 ab	16.5 ± 4.6 a	2.6 ± 1.0 ab	8.4 ± 2.9 a
	FL1	31.5 ± 2.2 ab	18.0 ± 2.9 a	3.0 ± 0.7 ab	6.8 ± 1.7 a
Peripheral	FL2	24.5 ± 2.2 ab	12.5 ± 3.6 a	2.1 ± 0.5 a	4.9 ± 0.2 a
	SLT	30.0 ± 2.2 ab	14.5 ± 2.2 a	2.6 ± 0.2 ab	6.5 ± 0.7 a
	ABM	35.5 ± 2.1 b	27.0 ± 2.5 a	5.1 ± 0.7 b	5.8 ± 0.7 a
	RSC	18.5 ± 2.8 a	12.5 ± 3.4 a	1.4 ± 0.4 a	6.3 ± 1.4 a
	WSD	28.5 ± 4.5 ab	16.0 ± 7.8 a	2.1 ± 0.6 a	7.8 ± 2.6 a
*p*	substrate	0.012	0.090	0.013	0.717

Germination percentage (GP), germination energy (GE), germination value after Czabator (GV_Cz_), and mean time to reach 50% of germinated seeds (MGT_50_). Different letters in the same column indicate significant differences among substrates. FL1 and FL2, soils from farmlands; SLT, soil from slates; ABM, soil from argillaceous blue marls; RSC, red sandy clay soil; WSD, forest soil on white sandy dunes.

**Table 6 plants-11-01981-t006:** Mean values (±SE) and significance level (*p*) of the germination parameters assessed in the greenhouse and under natural light, after different storage periods from collection, using seeds collected in June of each year. The test was carried out in July 2008.

Seed Type	Collection Year	GP (%)	GE (%)	VG_Cz_	MGT_50_ (Days)
	2008	16.0 ± 2.8 c	16.5 ± 2.4 b	0.5 ± 0.2 c	8.5 ± 0.5 a
Central	2007	33.8 ± 7.3 b	25.0 ± 7.9 b	4.2 ± 0.8 b	9.4 ± 1.2 a
	2006	62.8 ± 4.4 a	45.0 ± 6.2 a	22.3 ± 6.3 a	8.5 ± 0.3 a
	2005	71.1 ± 4.3 a	50.0 ± 6.2 a	24.7 ± 1.2 a	7.2 ± 0.2 a
	2008	16.0 ± 4.0 b	14.0 ± 4.7 b	0.5 ± 0.3 b	11.9 ± 2.7 a
Peripheral	2007	29.3 ± 4.5 a	23.0 ± 7.5 a	1.5 ± 0.6 ab	10.8 ± 0.8 a
	2006	26.3 ± 3.8 a	18.0 ± 2.2 ab	2.1 ± 0.6 a	6.1 ± 2.1 a
	2005	30.0 ± 4.2 a	19.0 ± 1.9 a	2.2 ± 0.5 a	6.3 ± 0.7 a
Significance	Year	<0.001	0.009	<0.001	0.747
level (*p*)	Seed type	0.005	0.016	<0.001	0.742
	Y × S	0.006	0.024	<0.001	0.007

Germination percentage (GP), germination energy (GE), germination value after Czabator (GV_Cz_), and mean time to reach 50% of germinated seeds (MGT_50_). Different letters in the same column indicate significant differences among collection year.

**Table 7 plants-11-01981-t007:** Mean values (±SE) and significance level (*p*) of the germination parameters assessed in the greenhouse and under natural light, one month after collection and using seeds collected in June 2008.

Seed Type	Treatment	GP (%)	GE (%)	VG_Cz_	MGT_50_ (Days)
Central	H_2_O_d_	16.0 ± 2.8	16.5 ± 2.4	0.5 ± 0.2	8.5 ± 0.5
	GA	94.0 ± 2.6	82 ± 4.7	14.7 ± 1.3	12.2 ± 0.2
Peripheral	H_2_O_d_	16.0 ± 4.0	14.0 ± 4.7	0.5 ± 0.3	11.9 ± 2.7
	GA	77.0 ± 7.7	69 ± 5.9	10.3 ± 1.7	11.4 ± 0.3
Significance	Treatment	<0.001	<0.001	<0.001	0.280
level (*p*)	Seed type	0.099	0.122	0.062	0.385
	T × S	0.099	0.281	0.060	0.183

Germination percentage (GP), germination energy (GE), germination value after Czabator (GV_Cz_), and mean time to reach 50% of germinated seeds (MGT_50_). Germination conditions: H_2_O_d_, distilled water; GA, gibberellic acid at 500 mg L^−1^ in H_2_O_d_.

**Table 8 plants-11-01981-t008:** Mean values (±SE) of seed fresh mass (mg) by seed type and collection year. The central seeds without the pappus but the peripheral ones with the attached phyllaries.

Seed Type	Collection Year		Total Average
	2005	2006	
Central	0.62 ± 0.01	0.58 ± 0.02	0.60 ± 0.01
Peripheral	2.84 ± 0.08	2.60 ± 0.19	2.72 ± 0.11

**Table 9 plants-11-01981-t009:** Mean values (±SE) and significance level (*p*) of the morphological properties of 8-day-old plants.

Seed Type	FM_p_ (mg)	LA_p_ (mm^2^)	DM_p_ (mg)	SLA (m^2^ kg^−1^)
Central	13.2 ± 1.4	39.7 ± 3.6	0.65 ± 0.07	61.5 ± 1.1
Peripheral	21.8 ± 1.6	53.3 ± 4.5	1.18 ± 0.09	44.9 ± 1.3
Significance level (*p*)	0.003	0.040	0.001	<0.001

Plant fresh mass (PM_p_). Plant leaf area (LA_p_). Plant dry mass (DM_p_). Specific leaf area (SLA).

**Table 10 plants-11-01981-t010:** Mean values (±SE) of chlorophyll content on fresh mass, dry mass and leaf area basis of *Picris willkommii* leaves. Sample size = 4. Leaves of 15–22 cm^2^ were used.

Trait	Chl*a*	Chl*b*
Fresh mass basis (mg g^−1^)	1.47 ± 0.30	0.36 ± 0.07
Dry mass basis (mg g^−1^)	16.98 ± 0.51	4.12 ± 0.33
Leaf area basis (mg m^2^)	490.4 ± 110.8	118.6 ± 24.6

**Table 11 plants-11-01981-t011:** Mean values at the end of the mineral nutrition and substrate trials, both without and with additional fertilization.

Treatment	RD (mm)	Flowers (Number)	DMap (g)	DMr (g)	DMap/DMr
Without additional fertilization					
Complete	4.50 b	16.8 b	1.84 b	0.93 b	2.02 ab
–N	1.50 a	2.5 a	0.08 a	0.11 a	0.93 ab
1/10N	1.50 a	3.5 a	1.18 a	0.13 a	1.38 ab
–P	1.33 a	1.5 a	0.11 a	0.06 a	2.67 b
–K	4.67 b	20.3 b	1.72 b	0.98 b	1.97 ab
–Mg	4.17 b	17.5 b	1.66 b	0.87 b	2.10 ab
–S	5.17 b	18.7 b	1.69 b	1.12 b	1.70 ab
–Ca	3.83 b	18.5 b	1.75 b	0.89 b	2.12 ab
–Fe	4.50 b	16.7 b	1.51 b	0.88 b	1.87 ab
FL1	1.00 a	1.8 a	0.08 a	0.26 ab	0.47 a
FL2	1.50 a	2.0 a	0.12 a	0.13 a	1.28 ab
SLT	1.50 a	2.5 a	0.11 a	0.09 a	1.42 ab
ABM	1.75 a	3.3 a	0.18 a	0.10 a	1.96 ab
RSC	1.75 a	2.8 a	0.16 a	0.21 ab	0.76 a
WSD	1.00 a	2.5 a	0.10 a	0.10 a	1.22 ab
*p*	<0.001	<0.001	<0.001	<0.001	0.006
With additional fertilization					
CompleteF	5.50 b	37.5 ab *	3.14 b *	1.02 ab	3.11 ab *
FL1_F_	5.50 b *	34.0 ab *	3.79 bc *	1.24 ab *	4.01 b *
FL2_F_	5.50 b *	56.0 b *	4.69 c *	3.09 b *	1.57 a
SLT_F_	3.50 a *	19.0 a *	1.70 a *	0.51 a *	3.36 ab *
ABM_F_	4.50 ab *	30.0 ab *	3.27 b *	1.42 ab *	2.29 a *
RSC_F_	5.00 ab *	27.5 ab *	2.49 ab *	1.06 ab *	2.37 a
WSD_F_	5.00 ab *	31.5 ab *	3.35 bc *	1.20 ab *	2.81 ab *
*p*	<0.001	<0.001	<0.001	<0.001	<0.001

RD, root diameter 0.5 cm below the root collar; DM: dry mass of aerial part (DMap) and root (DMr); *p*, significance level (one-way ANOVA). Different letters in the same column indicate significant differences among treatments within the same trial. Asterisks (*) indicate significant differences between the additional fertilization treatment and the one without additional fertilization (fertilized vs. unfertilized; *p* < 0.001 in all cases).

**Table 12 plants-11-01981-t012:** Mean values of the physical–chemical properties of the upper soil layer in the study area. FL1 and FL2, soils from farmlands; SLT, soil on slates; ABM, soil on argillaceous blue marls; RSC, red sandy clay soil; WSD, forest soil on white sandy dunes.

			Soil Type			
	FL1	FL2	SLT	ABM	RSC	WSD
pH	8.95	8.66	8.32	8.6	6.07	8.74
EC (dS/m)	0.26	0.29	0.13	0.21	0.27	0.22
OM (% *w*/*w*)	1.2	1.6	1.1	1.3	1.3	1.4
C/N	23.5	45.0	12.9	7.7	37.2	29.2
CO_3_ (%)	20.1	20.3	0.2	21.2	0.5	0.1
ESP (meq/100 g)	23.5	10.2	19.1	14.7	10.9	2.1
Active limestone (%)	10.0	3.84	0.10	6.32	0.11	0.0
N (mg kg^–1^)	304	202	493	962	209	278
P (mg kg^−1^)	28.2	19.6	33.6	39.4	40.8	31.3
K (meq/100 g)	0.7	0.1	0.1	0.2	0.2	0.05
Ca (meq/100 g)	33.0	38.3	7.1	40.3	3.8	7.9
Mg (meq/100 g)	4.7	6.2	6.1	5.7	2.6	0.8
Na (meq/100 g)	1.10	0.59	0.92	0.45	0.43	0.5
B (mg kg^–1^)	3.01	2.99	2.10	2.58	2.20	2.29
Fe (mg kg^–1^)	105.5	198.4	155.6	114.4	88.1	65.6
Cu (mg kg^–1^)	2.94	5.51	1.81	27.2	0.67	1.16
Zn (mg kg^–1^)	1.32	0.98	1.22	1.66	1.91	3.46
Mn mg kg^–1^)	64.5	67.9	115.2	60.5	2.70	8.18
Clay (%)	49.8	18.9	40.2	29.9	21.3	1.4
Silt (%)	31.8	3.4	13.2	15.8	7.7	0.4
Sand (%)	18.4	77.6	46.6	54.3	71.0	98.2
Texture (USDA)	Clay	Loamy sand	Sandy clay	Sandy clay loam	Sandy clay loam	Sand
Comments	Soil with active limestone. Low content of OM, N and P, but high of K, Ca and Mg.	Soil with active limestone. Low content of OM, N, P and K, but high of Ca and Mg.	Alkaline soil. Low content of OM and mineral nutrients, except for Mg.	Soil with active limestone. Low content of OM, N, P and K, but high of Ca and Mg.	Red sandy clay gravelly soil. Low content of OM and mineral nutrients.	Sand poor in mineral nutrients and OM. Low ESP.

EC, electric conductivity; OM, organic matter; ESP, exchangeable sodium percentage.

## Data Availability

Not applicable.

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
