# Peer review of "Seed Dormancy and Seedling Ecophysiology Reveal the Ecological Amplitude of the Threatened Endemism Picris willkommii (Schultz Bip.) Nyman (Asteraceae)"

_plants, 2022, doi:10.3390/plants11151981_

Round 1

Reviewer 1 Report

Plant growth is affected by many environmental factors. Thestudy aims to investigate the differences between the two seed types of P. willkommii ,a plant that lives  near the coast in the southwest of the Iberian Peninsula,Picris willkommii (Schultz Bip.) Nyman (Asteraceae) .The results are interesting.But the manuscript are preliminary and  does not have a clear data analysis, especially do not address the physiological or environmental effects of seed formation on gene expression in either species.Some measurement of some datas is puzzling. For example, net photosynthetic rate is obviously related to plant dry weight, but what is the relationship with germplasm heterogeneity? Water use efficiency is obviously related to rainfall, and the paper does not explain the rainfall.

Reviewer 2 Report

The authors investigated the seasonal pattern of life cycle, seed production, germination, plant emergence and development during three consecutive years in Picris willkommii. The authors assessed the growth phenology, the morpho-physiological traits of seeds, and the environmental factors (light, temperature, substrate). They found two types of seeds that can increase the probability of success.  Overall, this is a well-designed study and showed a huge amount of work. The findings should be value of plant conservation biology.

I only have two questions.

1. The work was carried out from 2006 to 2009. I'm curious about why the authors submit this work until now?

2. The most important part of the study is the two types of seeds of this species. The authors found that the central seeds showed significantly higher germination capacity and shorter germination time than the peripheral ones. It is interesting. I guess that it is likely because of the large peripheral seeds, which are easy to lose water and fall into deep dormancy, whereas the middle seeds are small, fall into shallow dormancy when they lose water, thus the germination rate is high and quick after absorbing water. This phenomenon is similar to the rules of the large seeds vs. small seeds among different species. Like K Strategist vs. r Strategist, such two Strategists within one species could cope with different environments and resources.  Please comment on this.

Round 2

Reviewer 1 Report

It is OK now.
